# A Michael Acceptor Analogue, SKSI-0412, Down-Regulates Inflammation and Proliferation Factors through Suppressing Signal Transducer and Activator of Transcription 3 Signaling in IL-17A-Induced Human Keratinocyte

**DOI:** 10.3390/ijms22168813

**Published:** 2021-08-16

**Authors:** A-Ram Kim, Seungbeom Lee, Jung U Shin, Seung Hui Seok, Young-Ger Suh, Dong Hyun Kim

**Affiliations:** 1Department of Dermatology, School of Medicine, CHA University, 120 Haeryong-ro, Pocheon 11160, Gyeonggi-do, Korea; tmfdkdjssl@hanmail.net; 2Department of Pharmacy, College of Pharmacy, CHA University, 120 Haeryong-ro, Pocheon 11160, Gyeonggi-do, Korea; lastchaos21c@snu.ac.kr; 3Department of Dermatology, CHA Bundang Mediacal Center, School of Medicine, CHA University, 59 Yatap-ro, Bundang-gu, Seongnam 13496, Gyeonggi-do, Korea; likesome@gmail.com (J.U.S.); a176058@chamc.co.kr (S.H.S.)

**Keywords:** STAT3, IL-17A, IκBζ, inflammation, proliferation

## Abstract

The activation of signal transducer and activator of transcription 3 (STAT3), as well as up-regulation of cytokines and growth factors to promote STAT3 activation, have been found in the epidermis of psoriatic lesions. Recently, a series of synthetic compounds possessing the Michael acceptor have been reported as STAT3 inhibitors by covalently binding to cysteine of STAT3. We synthesized a Michael acceptor analog, SKSI-0412, and confirmed the binding affinity between STAT3 and SKSI-0412. We hypothesized that the SKSI-0412 can inhibit interleukin (IL)-17A-induced inflammation in keratinocytes. The introduction of IL-17A increased the phosphorylation of STAT3 in keratinocytes, whereas the inactivation of STAT3 by SKSI-0412 reduced IL-17A-induced STAT3 phosphorylation and IκBζ expression. In addition, human β defensin-2 and S100A7, which are regulated by IκBζ, were significantly decreased with SKSI-0412 administration. We also confirmed that SKSI-0412 regulates cell proliferation, which is the major phenotype of psoriasis. Based on these results, we suggest targeting STAT3 with SKSI-0412 as a novel therapeutic strategy to regulate IL-17A-induced psoriatic inflammation in keratinocytes.

## 1. Introduction

Psoriasis is one of the most common inflammatory skin disorders, affecting from 2% to 3% of people worldwide [1]. The histopathology of psoriasis includes hyperproliferation of keratinocytes and increased infiltration of inflammatory cells, including T cells. The subsets of T cells that have been known as potential causes of psoriasis include Th1, Th17, and γδ T cells. These T cells induce psoriasis via the secretion of proinflammatory cytokines such as tumor necrosis factor-α (TNF-α), interleukin (IL)-17A, IL-23, and interferon-γ [2,3]. Over the last decade, the biological blockade of IL-17A has improved the symptoms of psoriasis significantly [4]. As a result, IL-17A has been the subject of much research as a major causative cytokine in the pathogenesis of psoriasis.

IL-17A has been known to play a key role in the transcriptional activation of the transcriptome in epidermal keratinocytes in psoriatic lesions. IL-17A activates the STAT3 in keratinocytes [1]. STAT3 has recently been revealed to play a role in the development and pathogenesis of psoriasis and psoriatic-inflammatory conditions [4,5,6,7]. STAT3 also functions as a transcription factor for inflammatory and immune responses through the phosphorylation of Tyrosine (Tyr) 705 followed by its dimerization and nuclear translocation [8,9]. The phosphorylation of STAT3 via receptor-interacting protein 4 (RIP4) causes the up-regulation of IL17A-mediated C-C Motif Chemokine Ligand 20 (CCL20) expression in keratinocytes [10], and RIP4 could be activated by the IL-17A-induced NF-κB signaling pathway in keratinocytes [11]. IL-6 and IL-22, which are potent STAT3 activators, are known for their synergistic action with IL-17A in the pathogenesis of psoriasis [12]. A recent study has shown that IL-17A-induced IL-6 causes the activation of STAT3, which is followed by increased cell proliferation [13]. IL-6-induces the phosphorylation of Tyr-705 STAT3, which increases the induction of Ki-67 as a cell proliferation marker in psoriatic keratinocytes [14]. Meanwhile, Janus Kinase (JAK) inhibitors have been determined to suppress the expression of NF-kappa-B inhibitor zeta (*NFKBIZ*, the encoding IκBζ), which is an essential IL-17A-mediated gene, via the regulation of STAT3 phosphorylation [5]. IκBζ affects the IL-17A-induced psoriatic genes in the keratinocytes of the psoriatic skin. S100 calcium binding protein A7 (S100A7) and human beta-defensin 2 (HBD-2, encoded by *DEFB4*) are representative antimicrobial peptides and proteins mediated by IκBζ, and the induction of these molecules causes the activation of immune cells via multiple mechanisms in psoriatic skin [15]. In addition, the multifunctional cytokines induced by IκBζ such as IL-6, IL-8, IL-17c, and CCL20 cause keratinocyte hyperproliferation and regulate the interaction with keratinocytes and immune cells. The inhibition of these cellular events is critical for the treatment of psoriasis [15,16,17]. Therefore, the inhibition of STAT3 was suggested as an additional therapeutic option, despite existing effective biological agents against IL-17 and IL-23. Treatment with Ochromycinone (STA-21), which is known as a small STAT3 inhibitor, showed approximately 75% improvement in the psoriatic phenotype, and the topical application of STAT3 inhibitor peptides showed anti-inflammatory effects that caused the down-regulation of IL-17A secretion in the imiquimod psoriasis mouse model [7].

In this study, we outline the synthesis of SKSI-0412 as a STAT3 regulator. SKSI-0412 suppressed the phosphorylation of STAT3 in IL-17A-induced keratinocytes. Subsequently, the inactivation of STAT3 repressed the transcription of NFKBIZ, and down-regulated the target genes of IκBζ (*S100A7, DEFB4*, and *IL-6*), although IL-17A was stimulated. S100A7 and HBD-2, known as antimicrobial peptides, are related to innate and adaptive immunity, and IL-6 affects keratinocyte proliferation in psoriasis. We also confirmed that SKSI-0412 affects cell proliferation via the regulation of IL-6-induced Ki67 expression. Our data showed that SKSI-0412 may offer the possibility of psoriasis treatment.

## 2. Results

### 2.1. Synthesis of SKSI-0412 (4)

The concise and convergent synthesis of the representative compound and derivatives is shown in Scheme 1. SKSI-0412 (4) was successfully synthesized via α-olefination of ketone 3, which was prepared via a tandem Claisen condensation/decarboxylation reaction sequence of ester 2 as a key step. Ester 2 was synthesized via the alkylation of phenol 1, followed by cyclization in N,N-diethylaniline at a high temperature.

### 2.2. Fluorescence-Based Analysis of SKSI-0412 (4) Binding to STAT3

We previously reported that 2-(4-(tert-butyl)phenyl)-1-(5-methoxy-2,2-dimethyl-2H-chromen-6-yl)prop-2-en-1-one (SLSI-1) directly binds to STAT3 and may form a covalent bond between the thiol moiety of STAT3 cysteine and the Michael acceptor of SLSI-1 [18,19]. In addition, we previously reported the dissociation constant (Kd value) for SLSI-1 binding to STAT3 [20]. In this study, the Kd values of SKSI-0412 (4) binding to STAT3 were obtained by the fluorescence-based binding titration experiment. At 340 nm, which is the maximum emission wavelength of STAT3, SKSI-0412 (4) exerted a moderate concentration-dependent inhibitory activity, which decreased the fluorescence intensities of STAT3 with a Kd value of 26.78 μM. This result implied that SKSI-0412 (4) more tightly binds to STAT3 than SLSI-1 (Kd = 43.13 μM) (Figure 1).

### 2.3. Inhibition of IL-17A-Induced STAT3 Tyr705 Phosphorylation via SKSI-0412 at Early Time Point

We investigated the role of SKSI-0412 in keratinocyte-like psoriasis. IL17-A is known as a key effector in psoriasis, and the phosphorylation of STAT3 contained with its expression was increased in psoriasis lesions. Therefore, we hypothesized that SKSI-0412 could affect the activation of STAT3 and act as a down-regulator of psoriatic biomarkers in keratinocytes induced by IL17-A. We have confirmed that treatment of IL17-A into keratinocytes induced an increase in the phosphorylation of STAT3 compared with the nontreated control group (Figure 2A,B). The activation of STAT3 in keratinocyte was stable without an increase in STAT3 expression at an early time point (Figure 2B). However, the phosphorylation of STAT3 showed a tendency that was dependent on the amount of STAT3 expression from 1 h after IL17-A treatment. In addition, the STAT3 in keratinocytes was activated by an increase in IL17-A (Figure 2A). To confirm the function of SKSI-0412 as an inhibitor of STAT3, various concentrations of SKSI-0412 were administered into IL17-A-induced keratinocytes. The results showed that the activation of STAT3 decreased upon administration of SKSI-0412 in a dose-dependent manner (Figure 2C–E).

### 2.4. Inhibition of IκBζ Induction through SKSI-0412 Treatment in IL-17A-Induced Keratinocytes

STAT3 is known to affect the transcription of IκBζ via the IL-22/JAK/STAT pathway. Therefore, we administered SKSI-0412 into the IL-22-induced keratinocytes. SKSI-0412 suppressed the activation of STAT3 and the induction of IκBζ by IL-22 treatment in keratinocytes (Appendix A). Recently, Muromoto et al. reported that phospho-STAT3 and IL-17A complementarily increased IκBζ induction [5]. We thought that SKSI-0412 could affect the transcription of NFKBIZ followed by psoriatic gene expression. Interestingly, the increase in NFKBIZ was not observed at 15 min after IL-17A treatment, although the levels of phospho-STAT3 increased (data not shown). The correlation of IκBζ and phosphor-STAT3 was observed from 1 to 6 h after IL-17A treatment (Figure 3A,B). The expression of STAT3 was maintained shortly after IL-17A-treatment then gradually increased from approximately 1 h onward. The up-regulation of STAT3 in keratinocytes caused a quantitative increase in phosphorylated-STAT3 and IκBζ. SKSI-0412 affected only the phosphorylation rate of STAT3 without regulating STAT3 expression. In addition, IκBζ was down-regulated by administration of SKSI-0412 (Figure 3C). In addition, the amount of IκBζ that translocated into the nucleus was increased in the IL-17A treatment group but was decreased in the group treated with both IL-17A and SKSI-0412 (Figure 3D). These results indicate that the activation of STAT3 was important for the transcriptional activity of NFKBIZ in the psoriatic signaling pathway via IL-17A or IL-22 and SKSI-0412 could function as a key regulator of IκBζ in the treatment of psoriasis.

### 2.5. Improvements in Cellular and Molecular Disease Biomarkers through SKSI-0412 in IL-17A-Induced Keratinocytes

Figure 3 shows that the administration of SKSI-0412 into IL-17A-induced keratinocytes caused a repression of IκBζ induction. IκBζ, a mediator of IL17-A-induced transcriptome activity, is known as a key regulator of specific psoriasis-associated genes that include DEFB4, S100A7, and IL-16. We hypothesized that the SKSI-0412 could repress the transcription of DEFB4 and S100A7 as regulators of immune cell activity through the down-regulation of the transcription for IκBζ. We confirmed that the low expression level of IκBζ via SKSI-0412 administration caused a decrease in HBD-2 and S100A7 levels, which are psoriasis-associated genes (Figure 4A–C).

IL-17A has been known to increase the proliferation rate of keratinocytes as well as the expression of antimicrobial peptides associated with psoriatic inflammation. IL-17A could cause an increase in cell proliferation by the activation of the IL-6/STAT3 signaling pathway [13]. It has also been reported that IL-6 is up-regulated by IκBζ and could induce Ki-67 as a cell proliferation marker. We first confirmed that IL-6 could increase the activation of STAT3 and the induction of Ki-67 in keratinocytes (Appendix A). SKSI-042 suppressed the phosphorylation of STAT3, and the expression of Ki-67 was subsequently decreased in IL-16-induced keratinocytes (Appendix A). In this paper, Ki-67 was used as a key marker of keratinocyte hyperproliferation that was regulated by the IL-17A-induced STAT3 signaling pathway. IL-17A treatment increased the expression of IL-6 (Figure 4D) and Ki-67 (Figure 4E,F) in keratinocytes. More importantly, SKSI-0412 administration inhibited Ki-67 production and cell proliferation induced by IL-17A (Figure 4E–G). These data indicate that SKSI-0412 regulated the cell proliferation of keratinocytes through the inhibition of STAT3.

## 3. Discussion

It has been known that STAT3 plays a role in Th17 cell biology [21,22]. The absence of STAT3 induces the termination of Th17 differentiation, and the constitutive activation of STAT3 causes an increase in IL17 producing cells [23,24,25]. In addition, the activation of STAT3 causes Th17 lymphocytes to secrete cytokines containing IL-6, IL-23, and IL-22 to keratinocytes [26,27,28,29]. Recently, a few researchers reported that the expression and activation of STAT3 are increased in epidermal keratinocytes as well as in Th17 cells of psoriasis lesions [6,7]. It has been reported that the introduction of IL-17A to keratinocytes causes STAT3 activation followed by induction of keratin 17 as one of the psoriatic genes [11]. In K5.Stat3C mice, the transgenic mouse model constituted the activation of Tyr-705 STAT3 in keratinocytes, which indicated that the psoriatic phenotype induced hyperkeratotic lesions and rete ridges [6]. The Ki-67, as a cell proliferation marker in psoriasis skin, is regulated by the IL6/STAT3 pathway [14]. In the IL-17A-induced keratinocytes, the inactivation of STAT3 causes the down-regulation of IκBζ as a psoriatic transcription key mediator [5]. Nevertheless, research regarding STAT3 as treatment for psoriasis has been few. It was recently confirmed that STA-21 (ochrimycinone), a STAT3 inhibitor that was developed with the purpose of being a treatment for human breast cancer, has the possibility of treating psoriasis. The docking model between the STAT3 protein and STA-21, identified by structure-based virtual screening, has predicted that STA-21 will bind to the Tyr-705 site of STAT3 and form a number of hydrogen bonds between Tyr-705 and nearby residues [7]. The activation of the STAT3 Tyr-705 site induces dimerization of STAT3 in the cytosol, and the dimer is translocated into the nucleus to act as a transcription factor [30,31,32,33,34]. Therefore, the inhibition of Tyr-705 could be a target in the treatment of disease containing the STAT3 signaling pathway.

In this study, we confirmed that SKSI-0412 directly bound dose-dependent to STAT3. We have recently reported that the growth inhibition of retinoblastoma cells by SKSI-0412, one of a complex series of anti-retinoblastoma screenings. SKSI-0412 showed less potent growth inhibition in retinoblastoma with a GI50 value of 4.03 uM. However, the effectiveness of SKSI-0412 on psoriasis has not been reported. The structure of SKSI-0412 was derived from the reported HIF-1α inhibitor SH48, which interacts with the cysteine 259 residue of STAT3. SH48 inhibited the Tyr-705 site of STAT3 via its Michael acceptor moiety (α,β-unsaturated carbonyl), but it does not influence the activation of upstream kinases such as Janus kinase 2, which is involved in STAT3 phosphorylation [19]. SKSI-0412 consists of a Michael acceptor, similar to SH48, and a structurally optimized chromene unit. The Michael acceptor moiety of SKSI-0412 has been predicted to bind to the cysteine 259 site of STAT3 and repress Tyr-705 phosphorylation and STAT3 translocation, leading to the induction of autophagy in MCF 10A ras in the breast cancer cell line. Similarly, SKSI-0412 treatment could also repress phosphorylation and nuclear translocation of STAT3 in IL-17A-induced keratinocytes at an early time point. In the current study, we showed that SKSI-0412 inhibited the activation and transcription of STAT3. An interaction between SKSI-0412 and STAT3 appeared to occur after SKSI-0412 treatment. SKSI-0412, which can be efficiently synthesized, consists of a flexible and tunable chemical structure that includes chromene pharmacophore, in contrast to STA-21. SKSI-0412 possibly binds to the STAT3 protein via covalent bonding of its Michael acceptor, which is not the case with STA-21.

IκBζ is an important regulator in the development of psoriasis and an important transcriptional activator that mediates the downstream effects of IL-17A. IκBζ is an important regulator of several psoriasis-related genes, including IL-17A downstream genes such as DEFB4, S100A7, and IL-6 [16]. STAT3, a substrate for JAK/Tyrosine kinase 2 (TYK2) kinase, was recently published as a transcription factor for NFKBIZ in IL-17A-treated keratinocytes [5]. The activation of the JAK/TYK2 pathway is known to be regulated directly via IL-22 rather than IL-17A in psoriatic keratinocytes. We have confirmed that treatment of IL-22-stimulated keratinocytes with SKSI-0412 could suppress STAT3 phosphorylation and IκBζ induction. In addition, we have confirmed that the expression of IκBζ was up-regulated in the IL-17A treatment group, whereas SKSI-0412 administration dose-dependently repressed IκBζ. The decrease in IκBζ expression due to SKSI-0412 causes decreased nuclear translocation, which subsequently causes decreased expression of the psoriasis-associated genes (DEFB4 and S100A7) in IL-17A-induced keratinocytes. HBD-2 and S100A7 are secreted from psoriatic keratinocytes and activate the innate immune system through various mechanisms to induce inflammation, which contributes to the pathogenesis of psoriasis [35]. IL-17A and STAT3 could induce the accumulation of IκBζ in keratinocytes via a different pathway. IL-17A and STAT3 maintain the stability and regulate the transcription activity of IκBζ mRNA, respectively. We have shown that SKSI-0412 could inhibit the induction of IκBζ despite IL-17A-induced stabilization of IκBζ. Therefore, STAT3 inactivation may be crucial in regulating the IL-17-induced inflammatory response (Figure 5).

It was recently reported that STAT3 regulates the transcriptional activity of Ki-67 through the IL-6/STAT3 pathway [14]. IL-6, one of the psoriatic cytokines, is increased by the IL-17A/IκBζ pathway. It was also reported that IL-17A activates the IL-6/STAT3 signal pathway followed by cell proliferation in hepatocellular carcinoma [13]. We hypothesized that SKSI-0412 represses the activation of STAT3 and IL-6-induced expression of Ki-67, which will cause the down-regulation of cell proliferation. This hypothesis was proven by three findings. First, we confirmed that IL-6 induces the activation of STAT3, and SKSI-0412 could suppress this pathway. Next, we confirmed that SKSI-0412 reduced the expression of IL-17A-induced IL-6 in keratinocytes. In addition, the administration of SKSI-0412 into IL-17A-induced keratinocytes suppressed Ki-67 expression and cell proliferation (Figure 5). 

The monoclonal antibodies of IL-17 (secukinumab and ixekizumab) are the mainstays in the treatment of moderate-to-severe psoriasis [36]. However, STAT3 inhibitors have the possibility of being an effective treatment option and alternative for psoriasis. This has been proven in this study, wherein we have shown that SKSI-0412, as a STAT3 inhibitor, suppresses the expression of IκBζ and its target genes in IL-17A-, IL-22-, and IL-6-treated keratinocytes. IκBζ has been known as the key driver that determines that psoriasis phenotype. Nfkbiz knockout mice have shown resistance to imiquimod (IMQ) and IL-23-induced psoriasis, but IL-17A knockout mice have shown only partial resistance to the IMQ-induced psoriasis phenotype [37]. These observations indicate that IκBζ has multiple induction pathways by other psoriasis-associated cytokines such as IL-22, TNF-α, or IL-36 in addition to IL-17A [16,38]. However, systemic treatment with a STAT3 inhibitor could cause unexpected side effects because STAT3 affects the transcription and regulation of various signal pathways, including apoptotic pathways. Therefore, other researchers have instead applied topical applications of STAT3 inhibitors, and the results indicated an improvement in approximately 75% of patients with psoriasis [7]. Therefore, further studies are necessary to compare the effects of a topical application of a STAT3 inhibitor to those of vitamin D3 or steroids, and histological assessment of the treated skin is required to confirm its benefits.

In summary, SKSI-0412 inhibited the phosphorylation, nuclear translocation, and transcriptional activity of STAT3 in IL-17A-induced keratinocytes. The inactivation of STAT3 caused the down-regulation of IκBζ (as a transcriptional activator) and psoriatic genes related to immune reactivity and cell proliferation. This finding suggests that the inhibition of STAT3 can induce the simultaneous inhibition of the IL-17A response in the keratinocytes of psoriatic lesions. Thus, SKSI-0412 may serve as a therapeutic target to cure psoriasis.

## 4. Materials and Methods

### 4.1. General Methods for Chemistry

Unless otherwise described, all commercial reagents and solvents were purchased from commercial suppliers and used without further purification. Tetrahydrofuran was distilled from sodium benzophenone ketyl. Dichloromethane, acetonitrile, triethylamine and pyridine were freshly distilled with calcium hydride. Flash column chromatography was carried out using silica-gel 60 (230-400 mesh, Merck, Kenilworth, NJ, USA) and preparative thin layer chromatography was used with glass-backed silica gel plates (1 mm, Merck, Kenilworth, NJ, USA). Thin layer chromatography was performed to monitor reactions. All reactions were performed under dry argon atmosphere in flame-dried glassware. 1H NMR and 13C NMR spectra were recorded on a Bruker Avance III HD (800MHz, with a 5 mm CPTCI CryoProbe) spectrometers (Billerica, MA, USA). Chemical shifts are provided in parts per million (ppm, δ) downfield from tetramethylsilane (internal standard) with coupling constant in hertz (Hz). Multiplicity is indicated by the following abbreviations: singlet (s), doublet (d), doublet of doublet (dd), triplet (t), quartet (q), multiplet (m) and broad (br). Mass spectra and HRMS were recorded on VG Trio-2 GC-MS instrument and JEOL JMS-AX (Tokyo, Japan), respectively. The purity of final product was confirmed by reverse phase high performance liquid chromatography (HPLC) (Shimadzu, Tokyo, Japan, LC-20AD liquid chromatograph, waters/sunfire^®^ TM C18 5um (4.6 × 150mm)) (water: acetonitrile = 1:7, rt = 6.4 min, >96%).

### 4.2. Methyl 2,2-Dimethyl-2H-Chromene-6-Carboxylate (2)

To a solution of methyl 4-hydroxybenzoate (500 mg, 3.29 mmol) in acetone, were added sodium iodide (591 mg, 3.94 mmol), potassium carbonate (908 mg, 6.57 mmol), PEG600 (0.5 mL) and 3-chloro-3-methylbut-1-yne (0.74 mL, 6.57 mmol) at room temperature under argon atmosphere. After completion of reaction monitored by thin layer chromatography (TLC), the reaction mixture was concentrated, dissolved with ethyl acetate, washed with water, dried over magnesium sulfate and filtered. Concentration of the organic phase afforded methyl 4-((2-methylbut-3-yn-2-yl)oxy) benzoate: 1H NMR (800 MHz, CDCl3) δ 7.95 (d, J = 8.8 Hz, 2H), 7.22 (d, J = 8.8 Hz, 2H), 3.87 (s, 3H), 1.68 (s, 6H). The residue was dissolved by N,N-diethylaniline (20 mL) without further purification and warmed up to 200 °C under argon atmosphere. After completion of reaction, the reaction mixture was concentrated by reduced pressure and purified by flash column chromatography (ethyl acetate: hexanes = 1:50). The desired ester (2) was obtained as product, 374 mg (52%): 1H NMR (800 MHz, CDCl3) δ 7.78 (dd, J = 8.4, 2.2 Hz, 1H), 7.65 (d, J = 2.2 Hz, 1H), 6.75 (d, J = 8.4 Hz, 1H), 6.32 (d, J = 9.9 Hz, 1H), 5.62 (d, J = 9.9 Hz, 1H), 3.85 (s, 3H), 1.43 (s, 6H).

### 4.3. 2-(3,4-Dimethoxyphenyl)-1-(2,2-Dimethyl-2H-Chromen-6-yl)Ethan-1-One (3)

To a solution of ester (2) (91 mg, 0.42 mmol) and 3,4-dimethoxyphenylacetic acid (122 mg, 0.62 mmol) in N,N-dimethylformamide (5 mL) were slowly added 1M tetrahydrofuran solution of sodium bis(trimethylsilyl)amide (1.66 mL, 1.66 mmol) at −10 °C under argon atmosphere. The reaction mixture was kept stirred at −10 °C. After completion of reaction, the reaction mixture was quenched with aqueous solution of ammonium chloride (5 mL) at −10 °C and concentrated by reduced pressure. The residue was dissolved by ethyl acetate, washed with brine and water, dried over magnesium sulfate, filtered and concentrated. Flash column chromatography (ethyl acetate: hexanes = 1:10) of the residue afforded the desired ketone (3), 110 mg (78%): 1H NMR (800 MHz, CDCl3) δ 7.78 (dd, J = 8.5, 2.2 Hz, 1H), 7.72 (dd, J = 8.5, 2.2 Hz, 1H), 7.65 (d, J = 2.2 Hz, 1H), 7.58 (dd, J = 8.9, 2.1 Hz, 1H), 7.47 (dd, J = 8.4, 2.0 Hz, 1H), 6.87 (d, J = 8.4 Hz, 1H), 6.30 (d, J = 9.9 Hz, 1H), 5.65 (dd, J = 13.9, 9.9 Hz, 1H), 4.13 (s, 2H), 3.96 (s, 3H), 3.93 (s, 3H), 1.43 (s, 6H).

### 4.4. 2-(3,4-Dimethoxyphenyl)-1-(2,2-Dimethyl-2H-Chromen-6-yl)Prop-2-en-1-One (4)

To a solution of ketone (3) (10.9 mg, 0.032 mmol) in N,N-dimethylformamide (1 mL), were added potassium carbonate (13.4 mg, 0.097 mmol) and paraformaldehyde (2.9 mg, 0.097 mmol) at room temperature under argon atmosphere. After completion of reaction, monitored by thin layer chromatography (TLC), the reaction mixture was concentrated, dissolved with ethyl acetate, washed with water, dried over magnesium sulfate, filtered and concentrated by reduced pressure. Flash column chromatography (ethyl acetate: Hexanes = 1:10) of the residue afforded the desired final product, 11.1 mg (99%): 1H NMR (800 MHz, CDCl3) δ 7.70 (dd, J = 8.5, 2.1 Hz, 1H), 7.61 (d, J = 2.1 Hz, 1H), 6.97 (d, J = 2.1 Hz, 1H), 6.94 (dd, J = 8.3, 2.1 Hz, 1H), 6.79 (m, 1H), 6.74 (d, J = 8.4 Hz, 1H), 6.30 (d, J = 9.9 Hz, 1H), 5.88 (s, 1H), 5.61 (m, 1H), 5.43 (s, 1H), 3.93 (d, J = 2.5 Hz, 2H), 3.86 (s, 3H), 3.84 (s, 3H), 2.02 (s, 1H), 1.44 (d, J = 2.6 Hz, 3H), 1.43 (s, 6H); 13C NMR (201 MHz, CDCl3) δ 196.54, 157.76, 149.34, 148.90, 147.85, 132.25, 131.10, 129.89, 129.85, 128.58, 121.61, 120.60, 119.75, 117.12, 116.07, 111.06, 110.29, 109.66, 77.68, 60.39, 55.89, 29.68, 28.53, 28.45, 21.04, 14.18; HR-MS (ESI) calcd for C22H22O4 (M) 350.1518, found 350.1509.

### 4.5. Procedure for Binding Titration Experiments Using Fluorescence Measurements

To examine the binding properties of SKSI-0412 (4) to STAT3, fluorescence-based equilibrium binding experiments were performed. All titration experiments were conducted at 25 °C using a Jasco FP 6500 spectrofluorometer (Easton, MD, USA). Recombinant human STAT3 protein (Abnova, cat no. H00006774-P01) was equilibrated with various concentrations of ligands before measuring fluorescence emission. Ligand stock solutions were titrated into a protein sample dissolved in phosphate buffer (pH 7.4). The protein was excited at 280 nm, and the decrease in fluorescence emission at 340 nm after ligand binding was measured as a function of ligand concentration. All titration data were fit to a hyperbolic binding equation to obtain KD values.

### 4.6. Cell Culture and Treatment

Human primary Keratinocyte cells were purchased from ATCC and were maintained in 5% CO2 at 37 °C in Dermal cell basal medium ATCC (PCS-200-030TM) with supplement kit. For all experiments, cells were seeded at a density of 1 × 10^6^ cell/10cm^2^ cell culture dish KC cells were cultured for 48 h and washed twice with PBS. Next, 100 ng/mL IL17-A (Peprotech, Korea) and various concentration SKSI-0412 (10, 20,30 nM) treated into KC culture dish adding Ca2+ 1.5mM. The KCs were harvested at each time point after PBS washing.

### 4.7. Cell Proliferation Assay

The proliferation of human primary keratinocyte cells was measured using EZ-cytox Cell viability assay kit (Daeillab Service, Seoul, Korea). Each cells were seeded at a concentration of 4 × 10^3^ cells in 96-well plates. Next, 100 ng/mL IL17-A (Peprotech) and various concentration SKSI-0412 (1, 10, 20 nM) treated into KC culture dish. After 0, 24, and 48 h culture, the Ez-cytox solution was added into each well and incubated at 37 °C for 2 h. The absorbance 459 nm was measured by spectrophotometry. All proliferation assays were performed in triplets.

### 4.8. Quantitative Real-Time PCR

Cells were harvested and total RNA was extracted using TRIZOL (sigma). cDNA synthesis was performed using M-MLV reverse transcriptase (promega, Madison, WI, USA) according to the manufacturer’s protocol. Relative gene expression was confirmed by ExicyclerTM (Bioneer, Korea) with Accu power^®^ GreenStarTM qPCR premix (Bioneer, Korea). The following primers were used: S100A7: 5′-CCAAGCCTGCTGACGATGA-3′ (forward) and 5′-GACATCGGCGAGGTAATTTGTG-3′ (reverse), DEFB4: 5′- GCTTGATGTCCTCCCCAGACT-3′ (forward) and 5′- CAGGATCGCCTATACCACCAAA -3′ (reverse), NFKBIZ: 5′- CGTCTTTCTCATGCTGTCACG -3′ (forward) and 5′- AAAGGTGAAGTTCAGCCAGGG-3′ (reverse), MKI67: 5′- CGTCCCAGTGGAAGAGTTGT -3′ (forward) and 5′- CGACCCCGCTCCTTTTGATA-3′ (reverse), GAPDH: 5′- ACCACAGTCCATGCCATCAC -3′ (forward) and 5′- TCCACCACCCTG TTGCTGTA -3′ (reverse). PCR reactions were started with an initial heating step at 95 °C for 5min followed by 40 cycles of 95 °C (30 s), 60 °C (30 s), and 72 °C (30 s). Relative mRNA expression levels were calculated by normalization to the reference genes GAPDH using the 2^−ΔΔCT^ method.

### 4.9. Western Blot Analysis

Cells were washed in PBS and lysed in RIPA buffer. Protein concentrations were determined with the bicinchoninic acid protein assay reagent (Pierce, Rockford, IL, USA) using bovine serum albumin (BSA) (Sigma Aldrich, St.Louis, MI, USA) as the standard. Protein lysates (25 μg) were separated on 15–8% sodium dodecyl-sulfate polyacrylamide gel electrophoresis and transferred to PVDF membranes (Millipore, Burlington, MA, USA). After blocking, the membranes were incubated overnight with indicated primary antibodies on a shaker at 4 °C. The following antibodies were used for Western blot analysis: anti-p-STAT3 (phosphor-STAT3 at Tyr-705) (Cell Signaling, Beverly, MA, USA), anti-STAT3 (Cell Signaling), anti-HBD-2 (Abcam, Cambridge, UK), anti-S100A7 (Abcam), anti-IκBζ (Gentex, Zeeland, MI, USA), and β-actin (Sigma). After washing in PBST, the membranes were incubated with secondary perosidase-linked IgG (1:3000, Abcam) for 1 h. Immunoreativity was detected by enhanced chemiluminescence (ECL kit, Bio-rad, Hercules, CA, USA) using the GF imageQuant LAS4000 (GE healthcare, Chicago, IL, USA).

### 4.10. Immunofluorescence (IF) Staining

The human keratinocytss cultured with IL-17A and SKSI-0412 were fixed in PCS containing 4% paraformaldehyde for 10 min and permeabilized in PBS containing 3% BSA (Sigma, St. Louis, MO, USA) and 0.1% Triton x-100 (Sigma) for 30min. The cells were incubated with 1:500 primary antibody in PBS containing 3% BSA and 0.1% TritonX-100 for 1 hr at room temperature. The cells were washed three times with PCS and incubated with secondary fluorophore-conjugated antibody (1:1000 Alexa Fluor 488 anti-rabbit, Chemicon, Temecula, CA, USA) for 30 min at room temperature. The cells were washed three rimes with PBS and then they were mounted using VECTASHIELD^®^ (Vector laboratories, Burlingame, CA, USA) with DAPI (Vector Laboratories) on coverslips. Fluorescence microscopy was performed on an Olympus LX71 fluorescence microscope (Olympus, Tokyo, Japan).

## Data Availability

Not applicable.

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
