# Peer review of "A Michael Acceptor Analogue, SKSI-0412, Down-Regulates Inflammation and Proliferation Factors through Suppressing Signal Transducer and Activator of Transcription 3 Signaling in IL-17A-Induced Human Keratinocyte"

_ijms, 2021, doi:10.3390/ijms22168813_

Round 1

Reviewer 1 Report

This is an interesting and important manuscript showing the possible therapeutic effects of SKSI-0412 in psoriasis. However, this manuscript needs some revision for publication.

1.IL-17RA/RC do not associate with JAKs. The authors should explain the precise mechanism how IL-17A activates STAT3 phosphorylation in Introduction.

2.STAT3 phosphorylation via Tyk2 is the signaling pathway induced by IL-22 rather than IL-17A. The transcription of IkBzeta through STAT3 is also caused by IL-22 while IL-17A rather stabilizes the mRNA, not the transcription itself.  

Thus the authors should examine if SKSI-042 may inhibit STAT3 phosphorylation induced by IL-22 and should show the results rather than that by IL-17A.

3.The authors should also examine if SKSI-042 may inhibit STAT3 phosphorylation induced by IL-6 and should show the results since inhibition of IL-6-induced Ki-67 expression may be caused by the mechanism above.

4.There are a lot of typographical and grammatical errors in the manuscript. This manuscript needs the revision by native English writers.

Author Response

Point 1: IL-17RA/RC do not associate with JAKs. The authors should explain the precise mechanism how IL-17A activates STAT3 phosphorylation in Introduction.

Response 1:

We agree with you and have incorporated this suggestion throughout our paper.

“IL-17A has been known to play a key role in the transcriptional activation of the transcriptome in epidermal keratinocytes in psoriatic lesions. IL-17A activates the signal transducer and activator of transcription 3 (STAT3) in keratinocytes [1]. ~ Treatment with Ochromycinone (STA-21), which is known as a small STAT3 inhibitor, showed approximately 75% improvement in the psoriatic phenotype, and the topical application of STAT3 inhibitor peptides showed anti-inflammatory effects that caused the down-regulation of IL-17A secretion in the imiquimod psoriasis mouse model [7].”

Please find above the introduction that has been modified.

Point 2: STAT3 phosphorylation via Tyk2 is the signaling pathway induced by IL-22 rather than IL-17A. The transcription of IkBzeta through STAT3 is also caused by IL-22 while IL-17A rather stabilizes the mRNA, not the transcription itself. 

Thus the authors should examine if SKSI-042 may inhibit STAT3 phosphorylation induced by IL-22 and should show the results rather than that by IL-17A.

Response 2:

We experimented that SKSI-0412 could inhibit the phosphorylation of STAT3 and induction of IκBζ in IL-22-induced keratinocytes. These results are indicated in supplementary figure 1.

Point 3: The authors should also examine if SKSI-042 may inhibit STAT3 phosphorylation induced by IL-6 and should show the results since inhibition of IL-6-induced Ki-67 expression may be caused by the mechanism above.

Response 3:

We confirmed that SKSI-0142 could suppress the activation of STAT3 and expression of Ki67 in IL-6-induced keratinocytes. Also, IL-6-treated cell proliferation was downregulated by SKSI-0412. These results are indicated in supplementary figure 2.

Point 4: There are a lot of typographical and grammatical errors in the manuscript. This manuscript needs the revision by native English writers.

Response 4: This manuscript has been edited by a professional scientific English language editing service.

Reviewer 2 Report

The authors have presented a paper depicting the effects of a synthetic analogue on the inflammatory and proliferative functions of keratinocytes via STAT3 pathway. The paper provides an interesting approach to a novel agent which might prove to be a promising factor in the treatment of various conditions, including psoriasis. However, several pitfalls can be identified in the manuscript, as follows:

Major points:

  • There are significant issues with the language and grammar, to the point that it renders some sentences unreadable and might transmit some inadvertently wrong or corrupt information.
  • In Figure 3, it is unclear what groups are referenced as having particular statistical correlations with the marked symbols (e.g. * and #). Although this practice has been applied in some papers, it is recommended that the authors find a clearer way to convey the which data series are significantly different or have a rejected null hypothesis.
  • A Figure where the signaling cascade in psoriasis is presented in the keratinocyte would be welcome so it is clear to the readers where and how SKSI-0412 acts and what the level of molecular involvement is, and whether signaling interferences are present.
  • Some short discussions regarding the pharmacological, toxicological, and biological properties or negative effects of SKSI-0412 might prove useful. Also, information on whether this compound was considered for other disease models or by other authors would be helpful in appreciating the novelty of this paper.

Minor points:

  • the abbreviation STAT3 should also be offered in the abstract, after the first iteration of the long-form definition.
  • several fields in the back matter were not filled in: "data availability", "conflict of interest", and "institutional review board statement".

Author Response

Point 1: There are significant issues with the language and grammar, to the point that it renders some sentences unreadable and might transmit some inadvertently wrong or corrupt information.

Response 1:

This manuscript has been edited by a professional scientific English language editing service.

Point 2: In Figure 3, it is unclear what groups are referenced as having particular statistical correlations with the marked symbols (e.g. * and #). Although this practice has been applied in some papers, it is recommended that the authors find a clearer way to convey the which data series are significantly different or have a rejected null hypothesis.

Response 2:

We confirmed figure 3 and fixed that.

Point 3: A Figure where the signaling cascade in psoriasis is presented in the keratinocyte would be welcome so it is clear to the readers where and how SKSI-0412 acts and what the level of molecular involvement is, and whether signaling interferences are present.

Response 3:

We insert molecular mechanism of SKSI-0412 in Figure 5.

Point 4: Some short discussions regarding the pharmacological, toxicological, and biological properties or negative effects of SKSI-0412 might prove useful. Also, information on whether this compound was considered for other disease models or by other authors would be helpful in appreciating the novelty of this paper.

Response 4:

On the 20th of May, 2021, We have reported that the growth inhibition of retinoblastoma cells by SKSI-0412, one of a complex series of anti-retinoblastoma screenings. SKSI-0412 showed less potent growth inhibition in retinoblastoma with a GI50 value of 4.03 uM. However, the effectiveness of SKSI-0412 on psoriasis has not been reported. In our experiments, the cell viability of keratinocytes was maintained in 0~100 nM concentration of SKSI-0412 (data not shown).

Regarding the negative effects of SKSI-0412, we have elaborated on the last paragraph of discussions part. Please refer to the paragraph below:

“However, systemic treatment with a STAT3 inhibitor could cause unexpected side effects because STAT3 affects the transcription and regulation of various signal pathways, including apoptotic pathways. Therefore, other researchers have instead applied topical applications of STAT3 inhibitors, and the results indicated an improvement in approximately 75% of patients with psoriasis. Therefore, further studies are necessary to compare the effects of a topical application of a STAT3 inhibitor to those of vitamin D3 or steroids, and histological assessment of the treated skin is required to confirm its benefits.”

Point 5: The abbreviation STAT3 should also be offered in the abstract, after the first iteration of the long-form definition.

Several fields in the back matter were not filled in: "data availability", "conflict of interest", "institutional review board statement"

  • Response 5: Thank you for your suggestion. We agree with you and have incorporated this suggestion throughout our paper

Round 2

Reviewer 1 Report

The authors well addressed the problems I pointed out and appropriately revised the article.